# Multiple Defect Classification Method for Green Plum Surfaces Based on Vision Transformer

Weihao Su, Yutu Yang *, Chenxin Zhou, Zilong Zhuang and Ying Liu

Jiangsu Co-Innovation Center of Efficient Processing and Utilization of Forest Resources, College of Mechanical and Electronic Engineering, Nanjing Forestry University, Nanjing 210037, China; suvi@njfu.edu.cn (W.S.); zhouchx@njfu.edu.cn (C.Z.); zzl0702@njfu.edu.cn (Z.Z.); liuying@njfu.edu.cn (Y.L.)
* Correspondence: yangyutu@njfu.edu.cn

**Abstract:** *Green plums* have produced significant economic benefits because of their nutritional and medicinal value. However, *green plums* are affected by factors such as plant diseases and insect pests during their growth, picking, transportation, and storage, which seriously affect the quality of *green plums* and their products, reducing their economic and nutritional value. At present, in the detection of *green plum* defects, some researchers have applied deep learning to identify their surface defects. However, the recognition rate is not high, the types of defects identified are singular, and the classification of *green plum* defects is not detailed enough. In the actual production process, *green plums* often have more than one defect, and the existing detection methods ignore minor defects. Therefore, this study used the vision transformer network model to identify all defects on the surfaces of *green plums*. The dataset was classified into multiple defects based on the four types of defects in *green plums* (scars, flaws, rain spots, and rot) and one type of feature (stem). After the permutation and combination of these defects, a total of 18 categories were obtained after the screening, combined with the actual situation. Based on the VIT model, a fine-grained defect detection link was added to the network for the analysis layer of the major defect hazard level and the detection of secondary defects. The improved network model has an average recognition accuracy rate of 96.21% for multiple defect detection of *green plums,* which is better than that of the VGG16 network, the Desnet121 network, the Resnet18 network, and the WideResNet50 network.

**Keywords:** vision transformer; *green plums*; deep learning; multiple defect detection





## 1. Introduction

*Green plums* are widely distributed in hills and sloping forests all over the world. They are rich in a large number of amino acids, vitamins, lipids, trace elements, and other nutrients, of which a variety of natural acids are important for human metabolism and have a rich nutritional and economic value [1]. *Green plum* sarcocarp is crisp and tender; it is thick, the core is small, and the taste is sweet and sour, so it is very popular among people. Not only is it unique in flavor, healthy, and appetizing, but it is also beneficial to human health.

With the improvement in people's living standards, their demand for high-quality fruits is also increasing. Consumers are more inclined to buy fruits without defects, and fruit product manufacturers are more inclined to choose high-quality fruits as raw materials. However, *green plums* are susceptible to diseases, insect pests, and knocks during their growth and production [2], resulting in different defects. Damage to the nutritional content and appearance of the product caused by defects will affect the market and price of the product. After picking, *green plums* are not easy to preserve, and they need to be sorted and selected as soon as possible. However, in China, the sorting of *green plums* is mainly carried out manually. The efficiency of manual sorting is low and the cost is high, which makes it impossible to sort a large amount of greengage in a short time.

Moreover, the sorting experience requirements for people are very high, and the sorting accuracy cannot be guaranteed. In addition, the efficiency and accuracy of manual detection are affected by human fatigue. These are the factors that cause quality problems in the secondary processing of *green plums*. To improve the economic value and nutritional value of *green plums* and their products, it is of great significance to carry out a variety of defect detection and classification processes on *green plums* that utilize high-level automation and intelligence. The main defects of *green plums* are divided into four categories: scars, rot, flaws, and rain spots. These defects will lead to quality and nutritional problems for *green plums* and their products. Therefore, before their sale and further processing, it is necessary to carry out defect detection on *green plums*, reject unqualified *green plums*, and classify *green plums*. This research focuses on the detection of the above four types of defects in order to achieve accurate identification of the main defects and other defects.

Computer vision technology is equivalent to the role of human vision in fruit and vegetable quality inspection. It perceives images, interprets and recognizes characters electronically, and provides information for quality grading and sorting machines. By combining machine vision and image processing with the advancement of computer technology, such systems have been applied in different fields of food engineering to accurately identify product characteristic defects in real time [3]. With the development of machine learning [4–7], researchers have applied machine vision [8] and deep learning to defect detection, making the non-destructive testing of fruit processing technology more efficient and accurate. The efficiency and accuracy of defect detection have been greatly improved through machine vision and deep learning. Yao et al. [9] developed a defect detection model based on You Only Look Once (YOLOv5) and optimized the network aiming at kiwifruit defects. This model can accurately and quickly detect defects in kiwifruit. The detection accuracy rate reached 94.7%, nearly 9% higher than the original algorithm. It only takes 0.1 s to process a single image, realizing real-time high-precision detection of kiwifruit defects. R. Nithya et al. [10] developed a computer-aided grading system for mango defect detection to classify high-quality mangoes. After training and testing the system using the publicly available Mango database, an accuracy rate of 98 percent was obtained. Huang et al. [11] used a multichannel hyperspectral imaging system for non-destructive testing of apple varieties. They achieved the best overall classification accuracy of 99.4% in the near-infrared and full-region spectral ranges, whose wavelengths range from 550–1650 nm. The multichannel hyperspectral imaging system provides more spatial–spectral information, and the non-destructive testing effect is excellent. In their research on *green plum* surface defect detection, Zhou et al. [12] and Zhou et al. [13] proposed a computer vision system for *green plum* surface defect detection based on the convolutional neural networks VGG16 and WideResNet50, respectively, which can detect the main defects of *green plums*. The average accuracy rates were 93.8% and 98.95%, respectively. Although the main defects of *green plums* can be accurately identified, each *green plum* may have more than one defect. The previous detection methods for *green plum* defects could only identify and output the main defects of the recognized *green plums* but couldn't identify other defects. According to the degree of impact of defects on production, from large to small, the defects of *green plums* are characterized by rot, flaws, scars, and rain spots. The production of different *green plum* products has different requirements regarding the defects of *green plums*. For example, *green plums* should have no rot or flaw defects to produce *green plum* wine. These defects indicate that the *green plums* have become moldy and contain a large number of microorganisms in their bodies. Such defective *green plums* damage the quality of green plum wine and pose a risk to human health. However, these defective *green plums* can be used as fertilizer after fermentation [14]. *Green plums* with only small scar defects can also be used to produce *green plum* wine to improve production efficiency. However, for *green plums* whose main defect is the scar, it is impossible to know whether it has other defects if there are small-scale rot defects on the surface. Additionally, this type of *green plum* still has food safety problems and cannot be used to produce *green plum* products. *Green plums* with milder scars and rain spots only have surface problems and no

internal necrosis, so they can still be used as raw materials for *green plum* wine, candied fruit, and plum powder; *green plums* without defects can be further sold or processed in the market. Therefore, it is necessary to carry out multi-defect detection on *green plums*, which can improve secondary production efficiency and the utilization rate of defective *green plums*.

Rain spot defects are the most common among the four types of defects in *green plums*. Although *green plums* with rain spot defects will not cause food safety problems, these affect the quality classification of *green plums*. The previous defect detection methods could only identify the main defects but could not judge whether other defects threatened food production safety in *green plums*. This makes it impossible to ensure that such *green plums* identified as having rain spot defects will not have safety problems and thus cannot be used as the raw material for producing *green plum* products in the next step. They can only be completely discarded due to food safety issues, which greatly reduces the economic value of *green plums*. This study used a deep learning method based on the vision transformer (VIT). Compared with the WideResNet50, Resnet18, and VGG16 models, the vision transformer network model has higher accuracy, added hazard degree analysis, and fine-grained detection abilities. Using a multi-defect detection scheme, it can identify all of the surface defects of *green plums*. This system can accurately detect major and minor defects in the output, enabling a more meticulous classification of defective *green plums*. Therefore, the precision and accuracy in identifying defects of green plums can be improved.

This study has the following innovations: (a) Aiming at the multi-defect identification problem of *green plums*, a defect identification network model based on the VIT network was proposed. (b) Compared with the single-defect classification processing of traditional data sets, this study's data set was processed using multi-defect classification. (c) After the MLP layer, the *green plum* defect risk level analysis layer and fine-grained detection link were added. The contribution of this study lies in the realization of a more detailed classification of *green plum* defect levels, the ability to accurately identify major defects and the remaining minor defects, and the output of the results of multiple defects. A new method for identifying multiple defects on a surface is proposed.

## 2. Materials and Methods

### 2.1. Data Collection and Processing

The dataset used in this study was a batch of *green plums* from Zhangzhou, Fujian, and 2799 RGB images of *green plums* were collected through visible light images. To simulate the real scene of actual production and inspection, this research transported *green plums* on a conveyor belt and collected images, as shown in Figure 1. A light gate from Yue Jiang Company (Hong Kong, China) was installed on the conveyor belt, and when the *green plums* were transported to the light gate, the conveyor belt stopped, and the acquisition device located above collected images. The acquisition system is shown in Figure 1. Viewed from the top down, the acquisition system's first equipment is the camera holder, the second equipment is the camera, the third equipment is the light source holder, the fourth equipment is the light source, the fifth equipment is the light gate, the sixth equipment is the *green plum* to be photographed, and the seventh is the conveyor belt. The entire image acquisition stage is in a closed lighting environment, and the material of the conveyor belt has a light-absorbing effect. The defective *green plum* is located on the conveyor belt, and the LED ring light source is used for supplementary light. The camera bracket can be adjusted to keep the camera at a fixed height, and the *green plum* should be rotated at random angles during the shooting process to obtain multi-angle *green plum* defect pictures.

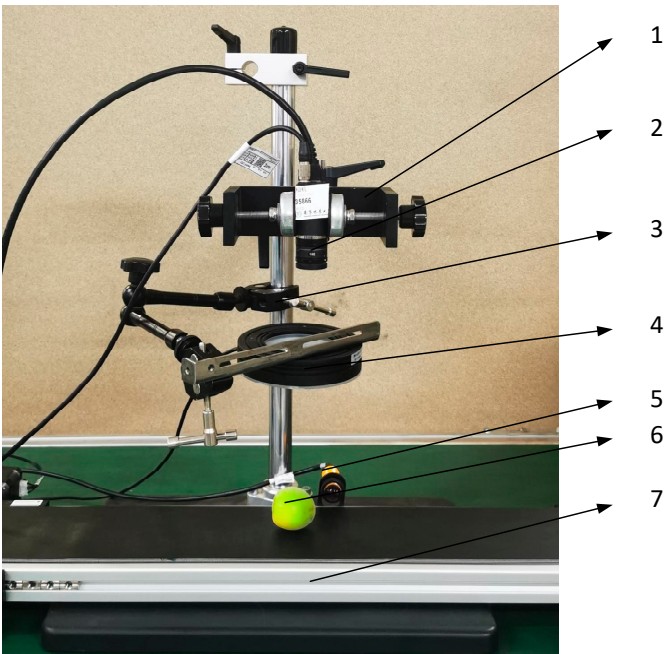

**Figure 1.** Collection equipment diagram: 1. camera holder; 2. camera; 3. light source holder; 4. light source; 5. light gate; 6. *green plum* sample; 7. conveyer belt.

The camera lens adopted is the M1620-MP2 industrial camera lens from Computer Company (Tokyo, Japan), whose focal length is 16 mm and minimum object distance is 20 cm. The industrial camera adopted the MER-531-20GC-P industrial camera of Beijing Daheng Image Technology Co., Ltd. (Beijing, China). A PYTHON 5000 frames exposure CMOS sensor chip was adopted. The light source used for the collection was an LED ring light source. The image collection stage was carried out in a closed lighting environment. During the shooting process, the *green plum* rotated to obtain multi-angle images of the surface defects of *green plums.*

In this study, 2799 pictures of various *green plum* defects and intact pictures were taken with a dot matrix camera, and the original pictures collected by the camera were 2592 × 2048 pixels. Due to the large size of the original image, in order to ensure the efficiency of image processing, the original images were preprocessed, and the noise in the image was removed at the same time [15]. The final image obtained had a size of 224 × 224 pixels. The defects of *green plums* were divided into four categories according to the degree of damage, from heavy to shallow: rot, flaws, scars, and rain spots. Among them, the rain spot defect had the characteristics of smallness, light color, and dispersion and occupied a small number of pixels in the image; thus, it is not easy to identify or misidentify [16]. At the same time, some plum pictures contained fruit stems from *green plums.* Although the feature of fruit stems is not a defect, it is affected by factors such as image acquisition angle, light changes, and lens distortion [17], resulting in the color and shape of fruit stems and rain spots. Consequently, the recognition of rain spots was disturbed. In Zhou H. Y.'s [12] previous *green plum* defect detection method, the algorithm (an improved VGG network model) did not yet solve the problem of misjudging fruit stems as defective rain spots. Traditional visual detection algorithms still have poor accuracy and limitations with fruit stems and rain spots [18], resulting in misjudgments of defects. Moreover, in the VIT model used by Zhang Xiao [19], compared to the recognition accuracy of other defects, the recognition error rate of rain spots was the highest, reaching 2.62%, which lowered the overall recognition accuracy. In order to avoid the misjudgment of rain spots and fruit stems and achieve higher recognition accuracy, the characteristic fruit stems were divided into one category for training. To sum up, *green plums* could be classified into

the following six categories: scars, rain spots, flaws, rot, intact, and fruit stems, as shown in Figure 2.

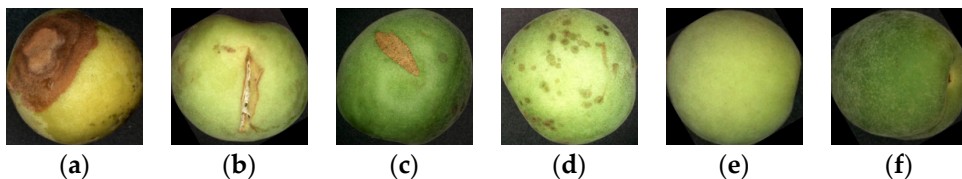

|  (a)  |  (b)  |  (c)  |  (d)  |  (e)  |  (f)  |

**Figure 2.** *Green plum* surface defect classification chart: (**a**) rot; (**b**) flaw; (**c**) scar; (**d**) spot; (**e**) intact; (**f**) stem.

*2.2. Dataset Processing Methods*

In terms of the classification method of the data set, the previous *green plum* defect research team chose to divide the *green plum* defects into four categories: rot, flaws, scars, and rain spots. When faced with *green plums* with multiple defects, they did not use the hazards of the defects as the classification standard. They chose the defect with the largest area as the defect class for the *green plums*. However, if a more harmful defect appeared in a small area, the final output could still be the defect in a larger area, ignoring the harm of other defects to the *green plums*. Moreover, small-area defects occupy fewer pixels, and training features may be lost after repeated convolution and pooling operations during training. This is also one of the reasons for the poor recognition effect of previous rain spot training. In contrast to the above classification methods, it was considered that in the actual detection process, multiple defects might appear on a *green plum*, as shown in Figure 3. In order to express and output the multiple defects of *green plums* more clearly, the dataset was divided more carefully. The sum of defects on each *green plum* picture was used as its defect category, as shown in Figure 3, which contains flaws, stems, and rain spots; then, this picture was used as the flaw + stem + spot category. In the actual detection, there were very few *green plums* with more than three kinds of defects, and some plum categories had only a few pictures or even none. In order to ensure the quality of the dataset, according to the hazard based on a combination of harmfulness and quantity, the following 18 theoretical types of combination classes were finally obtained: scar—1, scar + rot—2, scar + stem—3, scar + stem + spot—4, scar + spot—5, rot—6, rot + flaw—7, rot + stem—8, rot + stem + spot—9, rot + spot—10, intact—11, flaw—12, flaw + stem—13, flaw + stem + spot—14, flaw + spot—15, stem—16, stem + spot—17, spot—18, carried out with these types using order numbers 1–18 (Note: for concise expression, the category names and numbers in the following diagrams, such as the confusion matrix, correspond one-to-one). The combined images of 18 types of *green plum* defects are shown in Figure 4. They were divided into a training set, a test set, and a validation set. The images in the test set and validation set do not intersect. In addition, in order to ensure the quality of the dataset, data enhancement was performed on it, and operations such as mirroring, rotating, and adjusting the brightness and contrast of the original picture were performed. Finally, a total of 27,990 *green plum* sample pictures were obtained. The dataset was divided into a training set, a test set, and a validation set in the ratio of 8:1:1 and then enhanced. The category distribution of the dataset after image enhancement is shown in Table 1. Putting it into the VIT model, the VIT model could effectively learn various types of defect features and finally output all the defects of *green plums*. It only needed to classify them to meet the needs of improving the productivity of *green plums.* According to the degree of harmfulness of the defect, as long as the *green plums* with rot and flaws were listed as a hazard, as this type of *green plum* seriously affects food safety and the manufacturer can use it as fertilizer after fermentation, the *green plums* with scars and rain spots were listed as defective. Plums can be used as raw materials for secondary production. Fruit stems and perfect *green plums* are listed as normal plums, which can be further processed or sold directly.

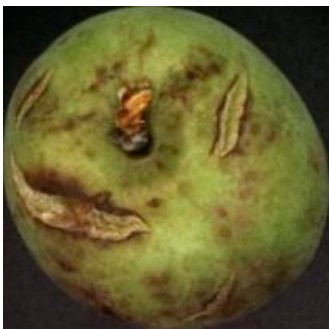

**Figure 3.** *Green plum* map with multiple defects (flaw + stem + spot—13).

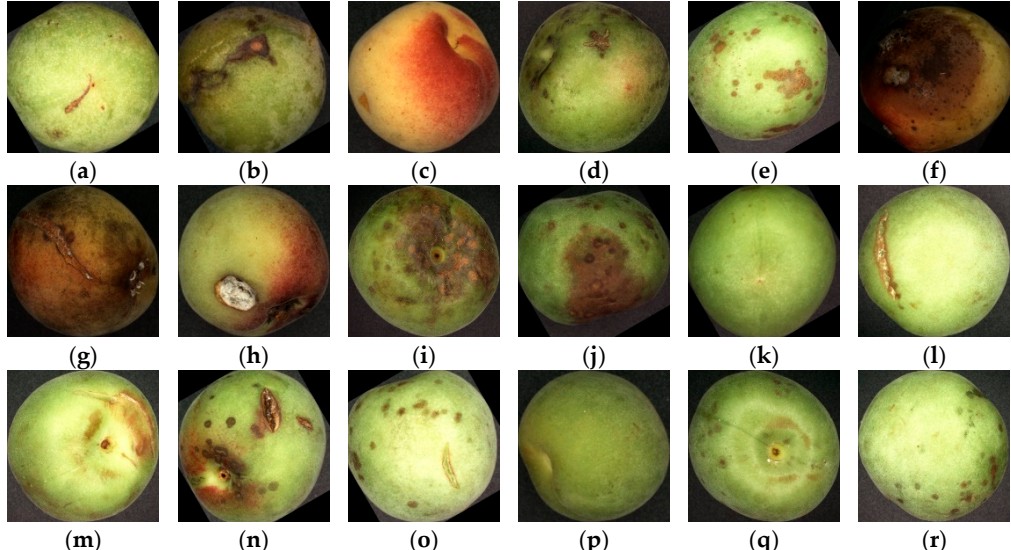

**Figure 4.** Shown are the 18 categories of *green plum* multi-defect classification: (**a**) scar; (**b**) scar + rot; (**c**) scar + stem; (**d**) scar + stem + spot; (**e**) scar + spot; (**f**) rot; (**g**) rot + flaw; (**h**) rot + stem; (**i**) rot + stem + spot; (**j**) rot + spot; (**k**) intact; (**l**) flaw; (**m**) flaw + stem; (**n**) flaw + stem + spot; (**o**) flaw + spot; (**p**) stem; (**q**) stem + spot; (**r**) spot.

### 2.3. Multiple Defect Detection Model of Green Plum Based on Vision Transformer

The vision transformer network [20] adopts self-attention and multi-head attention mechanisms, using residual connection and layer normalization techniques to accelerate training. The self-attention mechanism obtains information for each position in the input sequence. Among them, self-supervised learning reduces VIT's dependence on large-scale training [21]. Therefore, this study chose to use this network model to identify the *green plum* defects. In the field of image classification, the common convolutional neural network (CNN) [22,23] uses continuous stacking convolution layer operations to extract local features, which has certain limitations in extracting global features. As an encoder–decoder architecture based on the self-attention mechanism [24,25], the vision transformer model does not use RNN (cyclic neural network) sequential structure parallel training and can reflect complex spatial transformations and long-distance feature dependencies. Through the softmax function, the gradient is reduced. With the multiple sets of independent weights and parameter quantities added to the multi-head attention mechanism [26], the information obtained by different learning methods is combined, improving the expression ability of the network. Its global feature representation ability is stronger, and the migration effect is better.

**Table 1.** Distribution of dataset.

| Number | Class | Original Data Set | Data Augmentation | Validation Set | Training Set | Test Set |
|---|---|---|---|---|---|---|
| 1 | scar | 168 | 1680 | 168 | 1344 | 168 |
| 2 | scar + rot | 34 | 340 | 34 | 272 | 34 |
| 3 | scar + stem | 312 | 3120 | 312 | 2496 | 312 |
| 4 | scar + stem + spot | 37 | 370 | 37 | 296 | 37 |
| 5 | scar + spot | 18 | 180 | 18 | 144 | 18 |
| 6 | rot | 546 | 5460 | 546 | 4368 | 546 |
| 7 | rot + flaw | 62 | 620 | 62 | 496 | 62 |
| 8 | rot + stem | 130 | 1300 | 130 | 1040 | 130 |
| 9 | rot + stem + spot | 67 | 670 | 67 | 536 | 34 |
| 10 | rot + spot | 178 | 1780 | 178 | 1424 | 178 |
| 11 | intact | 616 | 6160 | 616 | 4928 | 616 |
| 12 | flaw | 114 | 1140 | 114 | 912 | 114 |
| 13 | flaw + stem | 62 | 620 | 62 | 496 | 62 |
| 14 | flaw + stem + spot | 30 | 300 | 30 | 240 | 30 |
| 15 | flaw + spot | 23 | 230 | 23 | 184 | 23 |
| 16 | stem | 54 | 540 | 54 | 432 | 54 |
| 17 | stem + spot | 60 | 600 | 60 | 480 | 60 |
| 18 | spot | 288 | 2880 | 288 | 2304 | 288 |

The VIT model consists of three modules: the linear projection of flattened patches (embedding layer), the transformer encoder, and the multilayer perceptron (MLP) head. The input image (224 pixels × 224 pixels) first passes through the embedding layer and is divided into 196 patches according to the size of 16 × 16. This step is realized by a convolution operation with a convolution kernel size of 16 × 16, a step size of 16, and a number of 768. The methods of adding position embedding and patch embedding can better reflect the information of the whole image. Secondly, the data enters the transformer encoder layer. The encoder contains a multi-head attention mechanism, which can represent the global features more accurately and repeatedly stack the encoder block L times. The output shape after the transformer encoder is consistent with the input shape. Finally, the defect classification result for *green plums* is obtained through the linear output in the MLP head [27]. Among them, the calculation formula for multi-head self-attention is as follows:

$$\text{MultiHead}(Q, K, V) = \text{Concat}(\text{head}_1, \cdots, \text{head}_h)W^O \tag{1}$$

where Q, K, V, H, and $W^O$ represent the query vector, key vector, value vector, number of heads, and output transformation matrix, respectively.

In Formula (1), the output $\text{head}_i$ of each head can be expressed as follows:

$$\text{head}_i = \text{Attention}\left(QW_i^Q, KW_i^K, VW_i^V\right) \tag{2}$$

In Formula (2), $W_i^Q, W_i^K$, and $W_i^V$ represent the query, key, and value transformation matrix of the $\text{head}_i$, respectively. The self-attention calculation formula is as follows:

$$\text{Attention}(Q, K, V) = \text{Softmax}\left(\frac{QK^T}{\sqrt{d_k}}\right)V \tag{3}$$

Although the classification method in this study could facilitate the network to learn various defects and output all the defects of *green plums*, it was also necessary to output the main defects according to the degree of defect damage. This study proposed primary defect detection based on previous research, enabling the new VIT network to output the first hazard defect of the *green plum* as the major defect based on the defect area size and hazard level. A hazard level analysis layer was added after the MLP output layer to obtain the main defects more accurately. In this layer, a convolutional neural network was inserted

into the network. It was used for training in the identification of the hazards of each defect. Different degrees and different types of defects have different effects on *green plums*. The network layer obtains the defect hazard factors of *green plums* and judges the degree of influence of the hazard on *green plums* through factor size analysis in order to determine the main defects of *green plums*. After analyzing the degree of harm, the entire network outputs the main defects of *green plums* more precisely. The main defect detection structure diagram is shown in Figure 5.

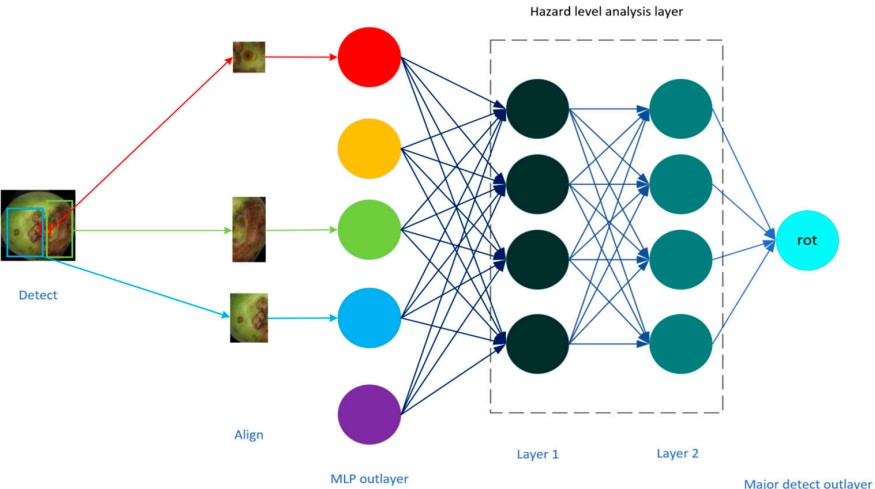

**Figure 5.** Major defect detection structure diagram.

The improved network could identify the most harmful defect features, but there may be multiple defects on the surface of *green plum*, causing the network to ignore the remaining defects and the harm they bring. In order to solve the multi-defect detection problem of *green plums*, the network structure was improved, and a fine-grained multi-defect detection link was added after the MLP. This link existed in parallel with the risk level analysis to identify all the defects for *green plums* and output them. In this link, multiple defects on *green plums* are first identified, the confidence of the corresponding defects in the graph is calculated, the confidence threshold is set to 0.6, and the confidence of defects higher than the threshold is output as secondary defects. If there are multiple secondary defects, the output sequence is in order of the degree of harm; finally, the network model can output all the defects in *green plums*, major defects + minor defects. The structure diagram of the whole network is shown in Figure 6.

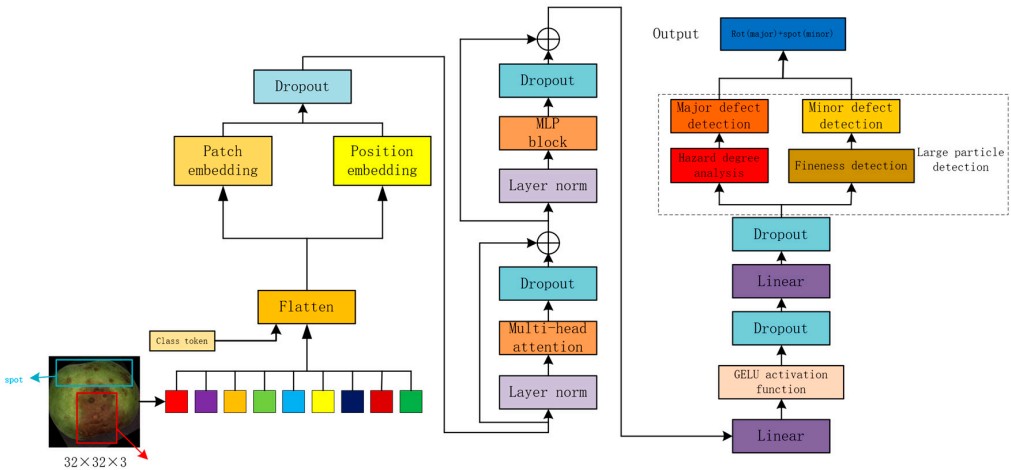

**Figure 6.** Network flowchart.

### 3. Results

The *green plum* defect classification network built in this research used the deep learning framework PyTorch to define the network calculation graph. The hardware, software, and compilation environment configurations used in this study are shown in Table 2.

**Table 2.** Software and hardware environment configuration.

| Software and Hardware | Name |
|---|---|
| System | Windows 10 × 64 |
| CPU | Inter I7 11700K@3.6 GHz |
| GPU | Nvidia GeForce RTX 3080Ti(12G) |
| Environment configuration | PyCharm 2022.3.3 + Pytorch 1.7.1 + Python 3.7.7 Cuda 10.2 + cudnn 7.6.5 + tensorboardX 2.1 |

Before training, the *green plum* defect classification network was parameterized. Batch_size was set to 64, Heads (the number of "heads" in the multi-head attention) was 2, Mlp_dim (the number of neurons in the hidden layer in the multilayer perceptron) was 64, and the learning rate parameter in the Adam optimizer was set to 0.01. After the parameters were set, the dataset was input into the model for training until the loss reached the minimum value and remained stable for 30 epochs. At this stage, the training of the *green plum* defect classification network model was completed. The *green plum* data from the test set were imported into the trained *green plum* defect classification network model. The model generated the test results for the main defects through the risk level analysis layer. As shown in Table 3, the accuracy rate of the VIT network for the classification of the main defects on the *green plum* surface reached 96.21%.

**Table 3.** Results of *green plum* defect classification.

| Methods | | Vision Transformer |
|---|---|---|
| | Scar | 94.02% |
| | Rot | 98.62% |
| Major Defect Classification Accuracy | Intact | 93.89% |
| | Flaw | 96.42% |
| | Spot | 93.68% |
| Accuracy | | 96.21% |
| Loss | | 0.078 |

Inputting 2799 test set pictures into the fine-grained defect detection link for testing, the confusion matrix of multi-defect detection on the *green plum* surface obtained by the VIT network is shown in Figure 7. The VIT model had the best detection effect on scar + spot and rot + stem + spot, with an accuracy of 100%. The effects of the intact category, the scar + stem + spot category, and the stem + spot category were poor. Among the 616 intact pictures, 1 was misjudged as the scar category, 4 were misjudged as the scar + stem category, and 15 were misjudged for rain spots. Among the 37 pictures in the scar + stem + spot category, 1 was identified as a spot, 1 was misjudged as a rot + stem, 1 was misjudged as a flaw + spot, and 6 were identified as a stem + spot kind. Among the 60 pictures of stem + spot, 4 were misjudged as scar + stem + spot, 1 was misjudged as a flaw, and 7 pictures were misjudged as spot without the characteristics of the stem.

Figure 8 is a test result diagram of a part of the test set, and the colored boxes in the figure show some misjudged plum cases. Picture (11 -> 18) in the green box in the figure misjudged the intact class as a spot. This may have occurred because the fruit tip of the intact *green plum* turned yellow, the rain spot was a small target defect, and its shape and color were similar to the fruit tip, resulting in misjudgment. The pictures in the purple frame (12 -> 13) identified the flaw as a flaw + stem category, which may have been caused by the fact that the pulp at the flaw was oxidized by air and was similar in color to the fruit

stems. The rain spot fruit stems in the yellow and red boxes are, respectively, identified as a rain spot and a flaw. The comprehensive analysis showed that because the rain spot defect was too dense, small, and round in shape, resulting in a misjudgment of the fruit stem, the rain spot defect was similar to a flaw when it was distributed laterally. The pictures in the blue box (1 -> 18) classified the scars as rain spots because the small scars were similar to rain spots, which led to their misjudgment.

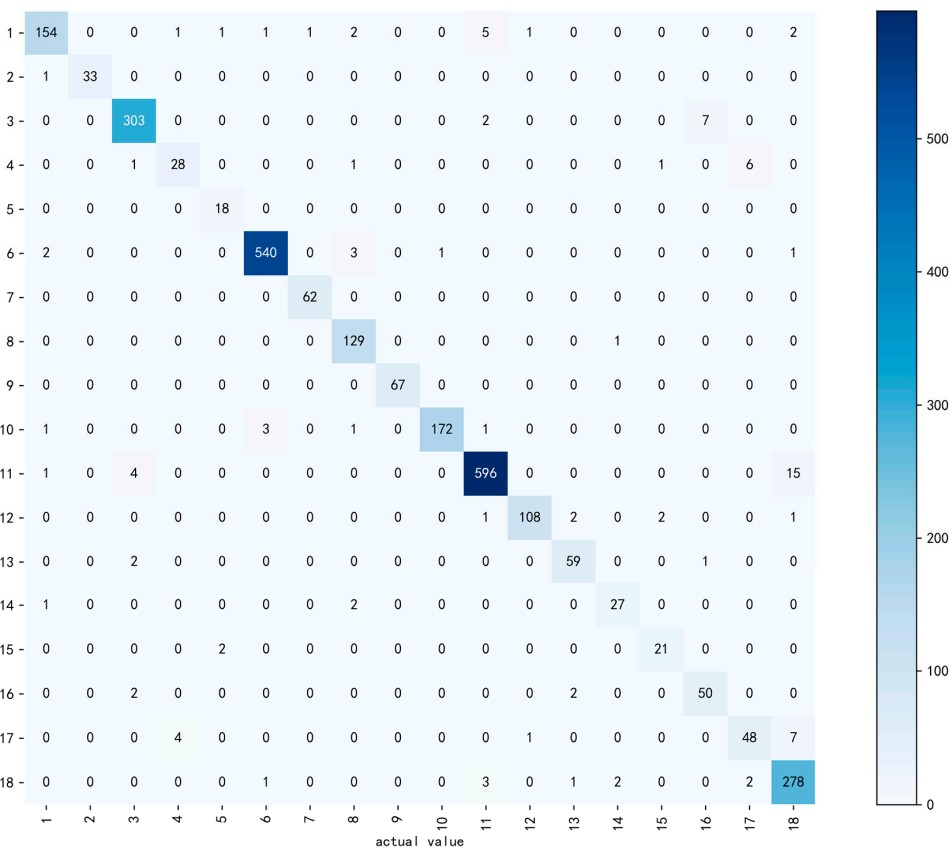

**Figure 7.** Confusion matrix. Type correspondence: scar—1, scar + rot—2, scar + stem—3, scar + stem + spot—4, scar + spot—5, rot—6, rot + flaw—7, rot + stem—8, rot + stem + spot—9, rot + spot—10, intact—11, flaw—12, flaw + stem—13, flaw + stem + spot—14, flaw + spot—15, stem—16, stem + spot—17, spot—18.

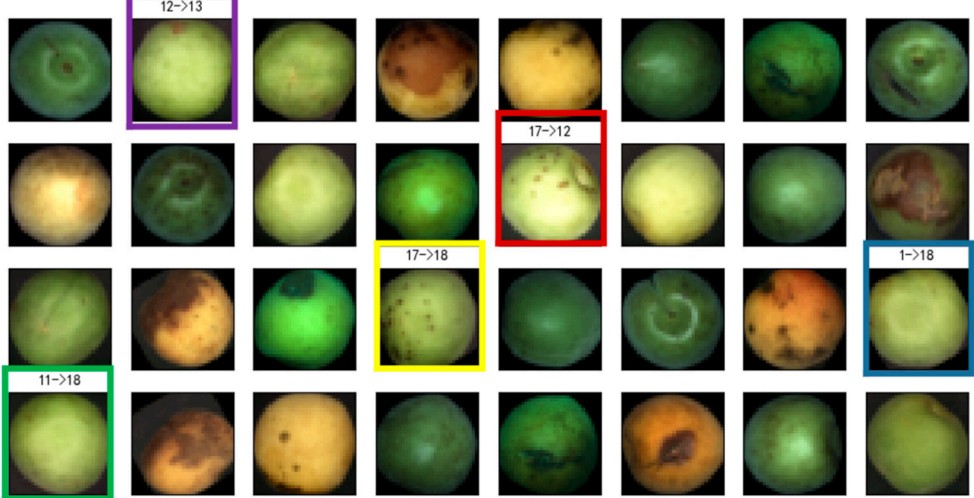

**Figure 8.** Results of the test.

## 4. Discussion

In this study, the multi-defect detection of *green plums* was a classification task. Compared with the target detection algorithm, the VIT model did not need to label each defect in each picture in the dataset. It only needed to classify different *green plum* defect pictures in the dataset. The target detection algorithm, such as the YOLO series model, needed to use labeling software to frame the defect area in each picture. Multiple defects often overlapped when labeling, resulting in repeated labeling of the frame, as shown in Figure 9. The yellow box in Figure 9 shows the stem, the blue box shows the scar, and the red box shows the rain spot. The rain spot feature and the rot feature overlapped. Furthermore, the process of manual labeling is a subjective job, after all, so there is also a certain error rate that will interfere with subsequent training. This not only consumes a lot of time but also leads to a decrease in recognition accuracy. Moreover, the purpose of this defect detection process was not to determine the exact position of the *green plum* [28], but only to identify the defect type of the *green plum*. Therefore, there was no need to label and locate defects in the dataset. Regardless of the perspective of dataset production or the final research goal, the target detection algorithm was unsuitable for this research.

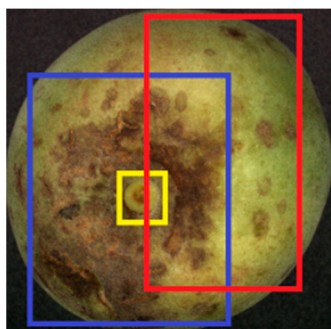

**Figure 9.** *Green plum* defect picture.

This study compared the performance of the *green plum* multi-defect classification network with that of other networks such as ResNet18, WideResNet50, Desnet121, and VGG16. This was conducted to validate the performance of the *green plum* network further. After the above models were fully trained, the accuracy rate of the main defect classification and the average test time were used as performance index comparisons. The test results are shown in Table 4.

**Table 4.** Accuracy of surface main defect classification for *green plum*.

| Accuracy of Surface Defect Classification | | | | | | Accuracy | Average Test Time |
|---|---|---|---|---|---|---|---|
| Network Name | Scar | Rot | Intact | Flaw | Spot | | |
| ResNet18 | 86.54% | 90.95% | 79.04% | 94.18% | 93.10% | 89.92% | 0.88 ms |
| WideResNet50 | 89.53% | 89.68% | 94.48% | 86.03% | 88.51% | 91.39% | 1.05 ms |
| Desnet121 | 93.83% | 92.63% | 96.57% | 89.52% | 97.70% | 94.14% | 1.39 ms |
| VGG16 | 92.34% | 95.18% | 98.06% | 90.83% | 97.17% | 95.42% | 0.96 ms |
| Vision Transformer | 94.02% | 98.62% | 93.89% | 96.42% | 93.68% | 96.21% | 1.43 ms |

In Table 4, in the classification of main surface defects of *green plums*, the accuracies of the VIT model for the main defects of scars, rot, intact, flaws, and rain-spotted *green plum* images reached 94.02%, 98.62%, 93.89%, 96.42%, and 93.68%, respectively. The average discrimination accuracy rate of the network was 96.21%, and the processing time of a single image was 1.43 ms. The accuracy rate of all kinds of main defect discrimination was significantly better than other models, such as WideResNet50. The VIT model could also identify other defects in *green plums*.

In terms of model accuracy, compared with ResNet18 [29] and WideResNet50, the vision transformer had a larger lead in the accuracy of the *green plum* multi-defect detection task. The vision transformer was higher than ResNet18 and WideResNet50 by 6.29% and 4.82%, respectively, and slightly ahead of the Desnet121 and VGG16 models [30] (2.07% and 0.79%). The detection accuracy for scars, rot, and flaws was higher than that of other models, and the accuracy regarding intact and rain spots was slightly lower than that of Desnet121 and VGG16 [31]. The overall effect of VIT was better. However, in terms of image processing time, since a hazard level analysis layer and a fine-grained detection link were added to the model, the VIT model took 0.55 ms longer to process a single image than the fastest ResNet18 but could obtain a high recognition rate of detection of main defects and multiple defects.

The loss curves of the vision transformer, ResNet18, WideResNet50, Desnet121, and VGG16 networks are shown in Figure 10. Although the VIT model outperformed other models' accuracy, its convergence speed during training was obviously not as good as other models. It may be that the effect of the optimizer in the VIT model was not as good as that of other models. In subsequent studies, we may consider replacing the optimizer with one that is more suitable for the VIT model in order to promote earlier convergence of the model and improve the efficiency of training.

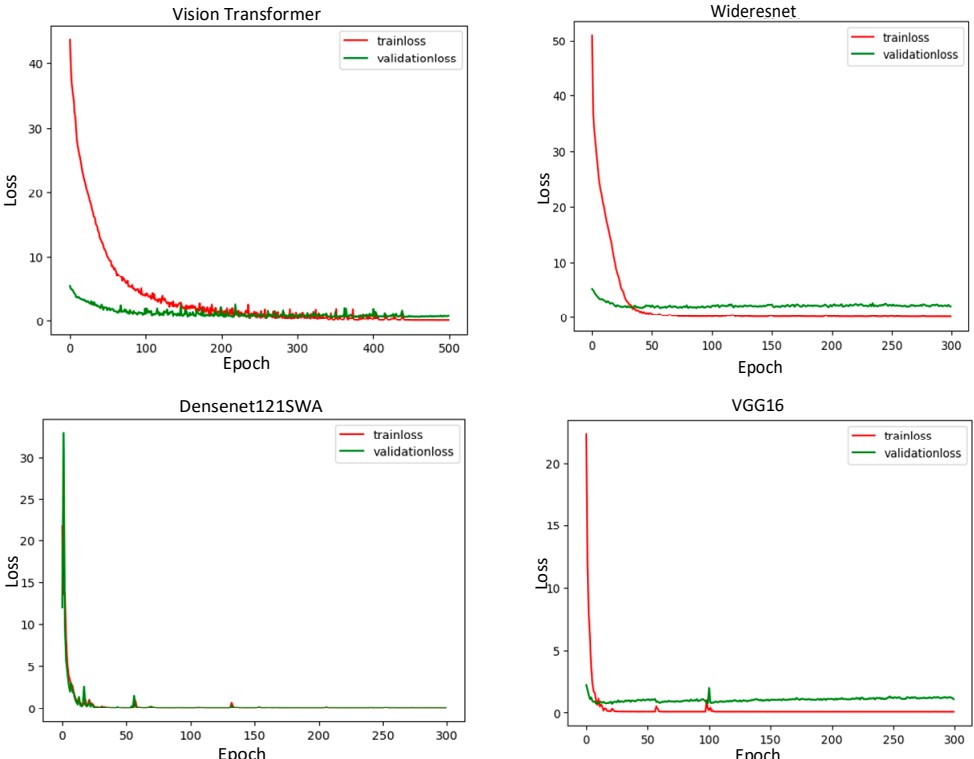

**Figure 10.** Loss curves of the four models.

Comprehensive analysis shows that the VIT network demonstrated excellent classification performance. To sort the quality of *green plums* based on the VIT network model and classify *green plums* with multiple flaws, the softmax function was used to reduce the gradient, and the multi-head attention mechanism was added. The overall feature representation ability is stronger, resulting in improved feature learning and migration effects. As a result, the network learns more features of defects, increasing the feature recognition rate. Consequently, the network performs better in the multi-defect classification of *green plums*. The average discrimination accuracy of the final model was 96.21%. This method not only accurately identifies the main defects of *green plums* but also classifies and outputs the defects in a more detailed manner and completes the multi-defect detection task of *green plums*. Manufacturers can consider the rational use of defective *green plums* that

can ensure food safety according to the defect situation of *green plums*. They can further classify multi-category *green plums* according to their own needs, which greatly improves the utilization rate of *green plums* and can increase the profit margins of enterprises.

## 5. Conclusions

Some researchers have previously conducted research on the defects of *green plums* and were able to identify the main defects. However, each *green plum* may have more than one defect. Their methods could only identify a single type of defect due to the lack of a detailed classification of *green plum* defects. The previous *green plum* defect detection methods could not detect other defects in *green plums*. This study proposed a model for the detection and classification of *green plum* multi-defects based on the vision transformer, aiming at the problem of multiple defects on the surfaces of *green plums*. This vision transformer was based on the four main defects of *green plums* (scars, rot, flaws, and spots) and a class of features (stems). There were 2799 *green plum* pictures classified with multiple defects to obtain a more detailed dataset, divided into 18 categories according to the actual situation. Moreover, the training set, validation set, and test set were allocated according to a ratio of 8:1:1. Then the dataset was expanded by changing parameters such as image angle, contrast, brightness, etc. to ensure the quality of the dataset and adding a risk level analysis layer and fine-grained detection links. The model was trained with the improved network. The network realized the effective classification of the main defects and multiple defects on the *green plum* surface, and the average recognition accuracy rate reached 96.21%. The single test image processing time was 1.43 ms.

This study also compared the established network with the accuracy of various major defects and the training loss curves of the ResNet18, Desnet121, WideResNet50, and VGG16 networks. The superiority of the vision transformer network was verified in defect classification performance compared to other network methods. It completed the automatic detection of multiple types of defects on the surfaces of *green plums* and classified the defect levels of *green plums* more carefully. However, there is still room for optimization in the training speed of the model. In addition, the more detailed classification method for *green plum* surface defects used in this study can also be applied to the defect detection of other fruits. This can help manufacturers further classify defective fruits and improve the utilization of non-hazardous and minimally hazardous fruits, thereby increasing production profit.

This research was based on static *green plum* surface images, and a static *green plum* surface multi-defect classification model was constructed based on the vision transformer model, achieving good surface multi-defect classification results. However, the training efficiency of the model was not high enough. This can be improved by changing the optimizer to accelerate the convergence speed of the model. The surface defect detection method used in this study could not understand the chemical composition of *green plums*, such as sugar content, pH, soluble solids, etc. It could not identify whether there were internal defects in the *green plums*. Moreover, under static conditions, only one side of the *green plums*' defects could be identified. In actual testing, the conveyor belt can be improved to make the *green plum* rotate continuously during transportation, allowing the camera to recognize all defects on the plum. In subsequent research, we should study how to identify internal defects in *green plums* in order to achieve higher food safety rates. Additionally, high-spectral imaging technology can be used to obtain the internal chemical components of *green plums* to select high-quality *green plums*.

**Author Contributions:** Conceptualization, W.S. and Y.Y.; data curation, W.S.; formal analysis, C.Z.; funding acquisition, Y.L.; investigation, Y.L. and Y.Y.; methodology, W.S.; project administration, Y.L.; resources, Y.Y.; software, W.S., Y.Y. and C.Z.; supervision, Z.Z.; validation, Y.L.; visualization, W.S.; writing—original draft, W.S.; writing—review and editing, Y.Y. and Y.L. All authors have read and agreed to the published version of the manuscript.

**Funding:** This research was funded by the Jiangsu agricultural science and technology innovation fund project (Project#: CX(18)3071): research on key technologies of intelligent sorting for *green plums*; and the 2020 Jiangsu graduate research and innovation plan (Project#: KYCX20_0882): research on *green plum* defect sorting systems based on artificial intelligence.

**Data Availability Statement:** Not applicable.

**Acknowledgments:** In addition to the funds we received, we would also like to thank Haiyan Zhou for providing us with materials and support for the image collection.

**Conflicts of Interest:** The authors declare no conflict of interest.

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
