# Peer review of "Multiple Defect Classification Method for Green Plum Surfaces Based on Vision Transformer"

_forests, doi:10.3390/f14071323_

Round 1

Reviewer 1 Report

Article has been written very well, structured properly as well
Title seems to be confusing, please change

Green plums are widely distributed in hills and sloping forests all over the world, rich
nutrients such as amino acids, vitamins, lipids, and trace elements. Many natural acids are important substances
for human metabolism, and have rich nutritional and economic values.

Authors took 2799 pictures of various green plum defects and intact pictures were
taken with a dot matrix camera, and the original pictures collected by the camera were 2592×2048 pixels and
green plums could be classified into the following six categories: scars, rain spots, flaws, rot, intact,
and fruit stems

Some areas numbers and units, must contain spaces, please check
Figures and Tables were adequate and with clarity
Conclusions are supported with data
Check for similar literature published, if any in other fruits

Author Response

Response to Reviewer 1

Comment 1: Title seems to be confusing, please change.

Response: We greatly appreciate your suggestion. We have changed the title to “Multiple defect classification method for green plum surfaces based on Vision Transformer”.

Comment 2: Check for similar literature published, if any in other fruits.

Response: Thank you for your suggestion. After the system search, we have added some literature: “Feature Reduction for the Classification of Bruise Damage to Apple Fruit Using a Contactless FT-NIR Spectroscopy with Machine Learning”, “Rapid and Non-Destructive Analysis of Corky Off-Flavors in Natural Cork Stoppers by a Wireless and Portable Electronic Nose”.

Reviewer 2 Report

The article uses a visual transformer network model to identify four types of defects in plums (scars, flaws, rain spots, and rot). To improve the differentiation between rain spots and stalks the stalks are identified in a separate class, and the different defects are combined and classified to determine the grade of plums and compared with other algorithms. The accuracy of the algorithm was verified. The following points are suggested for the authors' reference:
1, different positions of the plum image from the unit placement point photography may cause the defect of the missing, or each type should have the presence of the fruit stalk position point, this static image recognition method is still a certain distance from the real application.
2. Whether it is necessary to be able to recognize the size of some defects just by distinguishing the combination of different types of features to distinguish the grade, or whether the defects of small size will be recognized as another type of defects, thus leading to the problem of grade distinction. Or can only the rain spot and stalk features be recognized as excellent grades?
3. The number of parameters and computation of transformer models with the same accuracy often far exceeds that of CNN, and the current technology may be difficult to deploy them on edge platforms. So what is the superiority of this algorithm or the principle of selection?

Minor editing of English language required

Author Response

Response to Reviewer 2

Comment 1: Different positions of the plum image from the unit placement point photography may cause the defect of the missing, or each type should have the presence of the fruit stalk position point, this static image recognition method is still a certain distance from the real application.

Response: We greatly appreciate your thorough review. The collection method mentioned in the manuscript is primarily used in factory production lines. During the actual process of image collection, a stable illumination system was built, the green plum with multiple defects was rotated to take photos from multiple angles. During factory selection, the conveyor belt can be improved to make the green plum rotate continuously during transportation, allowing the camera to recognize all defects on the plum.

Comment 2: Whether it is necessary to be able to recognize the size of some defects just by distinguishing the combination of different types of features to distinguish the grade, or whether the defects of small size will be recognized as another type of defects, thus leading to the problem of grade distinction. Or can only the rain spot and stalk features be recognized as excellent grades?
Response: Generally speaking, the type and size of the defects on the picture are not related. Although they are all black defects, the decay is darker than the rain spot color, and the depth of the decay defect on the actual green plum will be deeper than the rain spot. Regardless of size, the effect of decay on the whole green plum is huge, and the large-scale rain spot is only the black spot on the surface of the green plum, which has little effect on the internal components and cannot be compared with the effect of decay defects. Fruit stems, as a feature of greengage, are not considered defects. Only the perfect and stemmed varieties can be considered excellent grades.

Comment 3: The number of parameters and computation of transformer models with the same accuracy often far exceeds that of CNN, and the current technology may be difficult to deploy them on edge platforms. So, what is the superiority of this algorithm or the principle of selection?

Response: The collection method mentioned in the manuscript is primarily used in factory production lines. In real factory sorting, GPUs with higher arithmetic power can be used for detection. VIT can achieve better detection results compared to CNN, and when used on edge devices, CNN models with smaller number of parameters and computational effort can be used.

Reviewer 3 Report

The research is very interesting and the 18 classes make the classification task difficult, but the result shows high test accuracy, which is excellent!

Line 31. Needs a reference.

Line 208. What are the differences between a test set and a verification set?

Line 287. Why the confidence threshold was set to 0.6?

Line 314. Please justify whether the 2799 picture is enough for the machine version task by summarizing the database size and other deep-learning articles in the area of agri-food science and computer science.

Line 325. Figure 7. Confusion matrix. Please clearly note the x and y axis, which one is the actual value, and which is the predicted value. A resolution needs improvement, at least 300 dpi.

Line 392. Figure 10. Seems more than 200 steps of training were not necessary, and would lead to over-fitting. Also, please clearly note the captions of the x and y-axis.

English is fine.

Author Response

Responses to Reviewer 3

Comment 1: Line 31. Needs a reference.

Response: Thank you for your suggestion. Reference has been inserted. We have cited “Future flavours from the past: Sensory and nutritional profiles of green plum (Buchanania obovata), red bush apple (Syzygium suborbiculare) and wild peach (Terminalia carpentariae) from East Arnhem Land, Australia”.

Comment 2: Line 208. What are the differences between a test set and a verification set?
Response: The images in the test set and validation set are composed of different defect green plum images, and there is no intersection between the two sets.

Comment 3: Line 287. Why the confidence threshold was set to 0.6?

Response: We experimented with several adjustments to the confidence level and found that the model was best when the confidence level was set at 0.6.

Comment 4: Line 314. Please justify whether the 2799 picture is enough for the machine vision task by summarizing the database size and other deep-learning articles in the area of agri-food science and computer science.

Response: In the area of agri-food science and computer science, 2799 pictures can already meet the needs of training: when the data set is insufficient, problems such as overfitting or gradient explosion will occur, but better results have been obtained on the test set. In most cases, larger data sets could achieve better results, and we will continue to expand this data set in future work.

Comment 5: Line 325. Figure 7. Confusion matrix. Please clearly note the x and y axis, which one is the actual value, and which is the predicted value. A resolution needs improvement, at least 300 dpi.

Response: We sincerely apologize for not painting a clear confusion matrix plot. and we have marked the actual value and predicted value on the X\Y axis, and increased the image resolution of the confusion matrix to 450dpi.

Comment 6: Line 392. Figure 10. Seems more than 200 steps of training were not necessary, and would lead to over-fitting. Also, please clearly note the captions of the x and y-axis.

Response: To prevent underfitting, we did not use early stopping in order to fully train the model. The model used for testing is the round with the highest validation set accuracy among all rounds. In Figure 10, the x/y axis has been defined as loss and epoch.

Reviewer 4 Report

See attached file.

Author Response

Response to Reviewer 4

Comment 1: Page 2, Lines 65-67: "With the development of machine learning, scholars at home and abroad have applied machine vision and deep learning to defect detection, and non-destructive testing fruit processing technology has 67 developed rapidly."

Please improve this sentence.

Response: We have made the following modifications: “With the development of machine learning, researchers have applied machine vision and deep learning to defect detection, making non-destructive testing of fruit processing technology more efficient and accurate.”

Comment 2: Page 2, Lines 69-71: "Yaojie [3] developed a defect detection model based on you only look once (YOLOv5), which can accurately and quickly detect defects in kiwifruit; the detection accuracy rate reached 94.7%, which is nearly 9% higher than the original algorithm."

I guess YOLOv5 is not designed for detecting defects in fruits. What do you mean by the detection accuracy rate reached 94.7%, which is nearly 9% higher than the original algorithm?

Response: We apologize for not expressing ourselves clearly. Aiming at the kiwifruit defects, they optimized the network by the following steps: (1) a small object detection layer is added to improve the model's ability to detect small defects; (2) we pay attention to the importance of different channels by embedding SELayer; (3) the loss function CIoU is introduced to make the regression more accurate; (4) under the prerequisite of no increase in training cost, we train our model based on transfer learning and use the CosineAnnealing algorithm to improve the effect.

Comment 3: Page 2, Lines 76-79: "Huang Y. P. et al. [5] used a multi-channel hyperspectral imaging system for nondestructive testing of apple varieties, and achieved the best overall classification accuracy of 99.4% in the near-infrared and full-region spectral ranges."

What do you mean by full-region spectral ranges?

Response: Full-region spectral ranges refer to the spectral range with wavelengths ranging from 550-1650 nm.

Comment 4: Page 2, Lines 92-96: "These defects indicate that the green plums themselves have become moldy and contain a large number of microorganisms in their bodies. Such defective green plums will not only damage the quality of green plum wine, but are also harmful to human health. This kind of green plum can be used as a fertilizer for planting after fermentation."

Can you cite the paper that has this claim?

Response: We have cited “A novel additional carbon source derived from rotten fruits: Application for the denitrification from mature landfill leachate and evaluation the economic benefits”.

Comment 5: Page 4, Figure 1: How realistic is this imaging setup? In real-life scenarios, the defect can be outside of the field of view of the camera. How can you guarantee that acquired images contain all defects of green plum?

Response: The collection method mentioned in the manuscript is primarily used in factory production lines. During the process of image collection, the green plum with multiple defects was rotated to take photos from multiple angles. In actual factory production, the conveyor belt can be improved to make the green plum rotate continuously during transportation, allowing the camera to recognize all defects on the plum.

Comment 6: Page 4, Lines 166-168: "Although the feature of fruit stems is not a defect, it is affected by factors such as image acquisition angle, light changes, and lens distortion [10], resulting in the color and shape of fruit stems and rain spots. Consequently, the recognition of rain spots was disturbed." Did you mean, in order to detect defects robustly, the visual analysis of green plums might be not enough?

Response: The use of stems is to describe the surface features of green plums in more detail. Stems are essential features on green plums, while defects are random features that appear on green plums, it is necessary to analyze them.

Comment 7: Page 5, Lines 208-209: "They were divided into a training set, a test set, and a verification set. "Did you mean training, test, and validation set?

Response: Yes, “verification set” has been modified to “validation set”.

Comment 8: Page 7, Lines 241-244: "The gradient is reduced through the softmax function, based on the multi-head attention mechanism [17], the information obtained by different learning is combined, and multiple sets of independent weights and parameter quantities are added to improve the expression ability." Please improve this sentence.

Response: We have improved the sentence to “Though the softmax function, the gradient is reduced. With the multiple sets of independent weights and parameter quantities added to the multi-head attention mechanism, the information obtained by different learning is combined, improving the expression ability of the network.”

Comment 9: Page 7, Lines 249-250: "It is realized by convolution, with a convolution kernel size of 16 × 16, a step size of 16, and several 768"

Please improve this sentence.

Response: We have improved the sentence to "This step is realized by convolution operation, with a convolution kernel size of 16 × 16, a step size of 16, and the number is 768."

Comment 10: Page 8, Lines 266-269: "This study proposed primary defect detection based on previous studies, enabling the network to output green plums’ first hazard defect as the primary defect, according to the size of the defect area and the degree of hazard." Which methods are you referring to? Please improve this sentence.

Response: We have improved the sentence to “This study proposed primary defect detection based on previous research, enabling the new VIT network to output the first hazard defect of the green plum as the major defect based on the defect area size and hazard level.”

Comment 11: Page 8, Lines 269-271: "To obtain the main defects more accurately, a hazard level analysis layer was added after the MLP output layer. In this layer, a convolutional neural network was inserted into the network." The network that you are showing in Figure 5 is not a convolutional neural network. It is a fully connected feed-forward neural network. What is the input of yellow and blue circles (nodes) in Figure 5?

Response: The input of yellow and blue circles (nodes) in Figure 5 is eigenvalue extracted by Convolutional neural network.

Comment 12: Page 14, Lines 446-449: "The surface defect detection method used in this study was unable to understand the chemical composition content in green plums, such as sugar content, pH, soluble solids, etc., and could not identify whether there were internal defects in the green plums. "Did you analyze hyperspectral images to have a better understanding of the chemical content of green plums?

Response: This is the direction of the collaborators' research, which will be reflected in future research.

Comment 13: Page 449-452: "Moreover, only the defects on one side of the green plum could be identified; in actual testing, defects that were not photographed on the other side of the green plum may have been ignored. It is necessary to rotate the green plum to ensure that all faces of the green plum can be detected." Although you partially answered my question number 5, the main challenge is to control lightning conditions in real-life situations in order to have a robust detection of green plum defects.

Response: The research mentioned in the manuscript is primarily applicable to factory production. In practical production, good lighting conditions can be achieved by adding a dark box and uniformly arranging light sources inside it.

Reviewer 5 Report

forests-2426809-peer-review-v1

The authors present an idea about using computer vision coupled with deep learning transfer learning to classify surface-defected green plums. The authors need to address the following comments.

Introduction:

The authors need to add some numbers about the performance of previous studies using either traditional or deep learning algorithms. This is critical to verify the novelty of the work

The main question is why shouldn’t we use a shallow or traditional machine learning algorithms if the problem is about binary classification?

Materials and Methods:

List any hardware as model (company, city, country).

Discussion:

Only 3 studies to compare your results with? This is not enough. Use studies with machine learning algorithms to compare with.

Figure 7: blurred and needs correction.

Figure 10. Fonts are not consistent. Make sure to get all fonts in figures in a consistent format.

Figure 10. Axis’ titles are not defined. Correct

Conclusions is way too long. Correct.

English language is fine

Author Response

Response to Reviewer 5

Comment 1: The authors need to add some numbers about the performance of previous studies using either traditional or deep learning algorithms. This is critical to verify the novelty of the work.

Response: We greatly appreciate your thorough review. In order to prove the originality of the article, references in deep learning have been added: “Feature Reduction for the Classification of Bruise Damage to Apple Fruit Using a Contactless FT-NIR Spectroscopy with Machine Learning”, “Rapid and Non-Destructive Analysis of Corky Off-Flavors in Natural Cork Stoppers by a Wireless and Portable Electronic Nose”.

Comment 2: The main question is why shouldn’t we use a shallow or traditional machine learning algorithm if the problem is about binary classification?
Response: Our classification is not a binary classification, but a multi-classification with a total of 18 categories. The surface texture features of greengage with defects are complex, and there are problems of complex color, size and distribution, and different defects will overlap with each other. Traditional machine learning needs to manually extract features, which is less robust and has not achieved good results. And using deep learning to apply to this scene can get better results.

Comment 3: List any hardware as model (company, city, country).

Response: We greatly appreciate your careful review and the model number of the hardware has been listed.

Comment 4: Only 3 studies to compare your results with? This is not enough. Use studies with machine learning algorithms to compare with.

Response: The surface texture features of greengage with defects are complex. Using machine learning to detect multiple defects requires designing a defect model for each defect. The deep learning model can update model parameters through backpropagation and detect multiple defects at the same time. Therefore, this article focuses on Deep learning network, comparing the detection effect of five deep learning network models.

Comment 5: Figure 7: blurred and needs correction.

Response: We sincerely apologize for not placing clear and explicit figures, and the image resolution of the confusion matrix has been increased to 450dpi.

Comment 6: Figure 10. Fonts are not consistent. Make sure to get all fonts in figures in a consistent format.

Response: Thank you for your careful review and all the fonts in figures has been modified.

Comment 7: Figure 10. Axis’ titles are not defined. Correct

Response: Thank you for your careful review and the X/Y axis has been defined as loss and epoch.

Comment 8: Conclusions is way too long. Correct.

Response: We greatly appreciate your thorough review and conclusions have been simplified.

Round 2

Reviewer 2 Report

I have no other comments on this manuscript.

Thank the authors for their corrections and replies.

Reviewer 4 Report

Although the authors did not completely answer all questions, the quality of the manuscript is much better than the previous version. I have no more comments.